# The MARC SE-Africa dashboard: Joining forces to counteract emerging antimalarial resistance in South and East Africa

Stephanie van Wyk[1,2,3,4]*, Ishen Seocharan[5,6], Eulambius M. Mlugu[1,2], Dhol S. Ayuen[3,4], Donnie Mategula[1,7,8,9], Tikhala Makhaza[1,7,8,9], James Kiarie[10], Victor Asua[1,11,12], Jimmy Opigo[13], Aimable Mbituyumuremyi[14], Kibor Kipkemoi Keitany[10], Emmah Mongina Nyandigisi[10], Pierre Sinarinzi[15], Peter Aguek Kon Baak[16], Tommy Nseka Manbul[17], Samwel Lazaro Nhiga[18], Sijenunu Aron Mwaikambo[18], Maulid Kassim[19], Sija Joseph Sija[19], Abdikarin Hussein Hassan[20], Michael Katende[21], Jaishree Raman[1,22,23,24], Karen I. Barnes[1,2,3,4]

**1** Mitigating Antimalarial Resistance Consortium for Southern and Eastern Africa (MARC SE-Africa), Cape town, South Africa, **2** Collaborating Center for Optimizing Antimalarial Therapy, University of Cape Town (CCOAT), Department of Medicine, Division of Clinical Pharmacology, South Africa, **3** WorldWide Antimalarial Resistance Network (WWARN), Oxford, United Kingdom, **4** Infectious Diseases Data Observatory (IDDO), Oxford, United Kingdom, **5** Medical Research Council of South Africa, Cape town, South Africa, **6** Biostatistics Research Unit, South African Medical Research Council, Durban, South Africa, **7** Liverpool School of Tropical Medicine, Liverpool, United Kingdom, **8** Malawi Liverpool Wellcome Programme, Blantyre, Malawi, **9** Kamuzu University of Health Sciences, Blantyre, Malawi, **10** National Malaria Control Programme of Kenya, Kenyatta National Hospital, Nairobi, Kenya, **11** Infectious Diseases Research Collaboration, Kampala, Uganda, **12** Institute of Tropical Medicine, University of Tubingen, Tubingen, Germany, **13** National Malaria Control Programme, Wandegeya, Kampala, Uganda, **14** National Malaria Control Programme Rwanda, Rwanda Biomedical Centre (RBC) or the Ministry of Health, Kigali, Rwanda, **15** National Malaria Control Programme of Burundi, Burundi Ministry of Public Health, Bujumbura, Burundi, **16** National Malaria Control Programme of South Sudan, Juba, South Sudan, **17** National Malaria Control Programme of the Democratic Republic of the Congo (DRC), Kinshasa, DRC, **18** National Malaria Control Programme, Dodoma, Tanzania, **19** National Malaria Control Programme of Zanzibar, Ministry of Health Zanzibar, Zanzibar, Tanzania, **20** National Malaria Control Programme of Somalia, Mogadishu, Banadir, Somalia, **21** East Africa Community Secretariat, Arusha, Tanzania, **22** Centre for Emerging Zoonotic and Parasitic Diseases, National Institute for Communicable Diseases, Johannesburg, South Africa, **23** Wits Research Institute for Malarial Research, University of Witwatersrand, Johannesburg, South Africa, **24** UP Institute for Sustainable Malaria Control, University of Pretoria, Pretoria, South Africa

* stephanie.vanwyk@gmail.com

## Abstract

Regions within eastern and southern Africa (SE-Africa) carry some of the highest malaria burdens. Understanding the spatiotemporal dynamics of the emergence and spread of artemisinin (partial) resistance (ART-R) and how to mitigate ART-R is therefore of paramount importance in these areas. Here, we present a dashboard developed by the Mitigating Antimalarial Resistance Consortium for SE-Africa in collaboration with nineteen national control malaria programs (NCMPs) and their partners. The dashboard supports NCMPs' decision-making by providing curated information on the latest available antimalarial resistance data. We systematically reviewed, collated, and visualized antimalarial resistance information from therapeutic efficacy studies, molecular surveillance for *Pfkelch13* ART-R genetic markers, current

**Data availability statement:** The MARC SE-Africa dashboard is embedded on a website and is a free-to-access tool available via https://www.samrc.ac.za/antimalarial-drug-resistance. All collated and generated data used for data visualization are accessible via https://tinyurl.com/AntimalRepo.

**Funding:** MARC SE-Africa (Grant Agreement n° 101103076) is supported by the Global Health EDCTP3 Joint Undertaking and its members. The funders had no role in study design, data collection and analysis, decision to publish, or preparation of the manuscript.

**Competing interests:** The authors have declared that no competing interests exist.

in-country malaria treatment policies, and reported malaria cases and deaths. We identified evidence gaps in therapeutic efficacy and molecular surveillance, particularly in southern Africa. Five countries, Angola, the Democratic Republic of Congo, Kenya, Tanzania and Uganda, reported artemether-lumefantrine treatment failures above the WHO threshold of 10% after correcting for reinfections. The A675V, R561H, P574L, and C469F *Pfkelch13* markers were highly prevalent in cross-border regions of several East African countries, with the C469Y marker rapidly spreading across Uganda. The dashboard provides an interactive platform for sharing regional data. We discuss the implications of these findings for policy, practice, and research.

## Author summary

We introduce the MARC SE-Africa Dashboard, an innovative tool developed at the request of national malaria control programs to enhance malaria management through data-driven insights into antimalarial drug resistance in Southern and Eastern Africa. This interactive dashboard compiles and visualizes data from multiple sources, offering an up-to-date overview of resistance patterns and treatment efficacy. With its interactive maps and user-defined parameters, the dashboard enables detailed multinational and cross-border analysis, which is invaluable for identifying and monitoring emerging hotspots of resistance. This allows health professionals, researchers, and policymakers to target interventions effectively and adapt strategies in response to the evolving landscape of malaria resistance. The dashboard visualizes updated information on molecular markers of resistance, therapeutic efficacy outcomes, and national treatment policies for 19 malaria-endemic countries. The dashboard aims to facilitate regional cooperation in malaria management, which is essential for proactive combat in high-risk areas. We discuss the insights gained from the curated and collated data and how this data and the dashboard can support national malaria control programs and researchers.

## Introduction

Malaria remains one of the world's most pressing public health concerns, where sub-Saharan Africa is disproportionately affected, accounting for 94% and 95% of the global cases and deaths, respectively [1,2]. Cases are increasing in East Africa, where antimalarial drug-resistant malaria has emerged independently and is spreading rapidly. Effectively tackling the substantial malaria burden and drug-resistant malaria in southern and eastern Africa (SE-Africa) requires data-driven, effective, and targeted interventions [3–5].

 Artemisinin-based combination therapies (ACTs) are the frontline treatment for uncomplicated *Plasmodium falciparum* malaria [1,6]. The effectiveness of this treatment modality is founded on the combined action of a fast-acting artemisinin derivative

with a longer-acting partner drug [7–9]. Over 20 years of extensive ACT use in Africa, particularly artemether-lumefantrine (AL), has, however, exerted selective pressure on *P. falciparum* parasite populations, driving the emergence and spread of artemisinin (partial) resistance (ART-R) [10]. ART-R is characterized by delayed parasite clearance following ACT treatment [5,11–13]. While ART-R alone does not lead to ACT treatment failure, it is the first step towards the development of multidrug resistance. This because the reduced effectiveness of the artemisinin component increases the pressure on the partner drug to eliminate residual parasites from infected patients [11]. Therefore, the spread of ART-R threatens the efficacy of ACTs and has a high likelihood of increasing malaria-related morbidity and mortality substantially in Africa. [14].

ART-R, first described in Southeast Asia in the early 2000s, has recently and independently emerged in East Africa [4–6,15]. A comprehensive understanding of the genetic and clinical determinants of ART-R remains invaluable to containing its spread [4–6,16]. For the past 10 years, molecular surveillance for non-synonymous mutations in the *P. falciparum* *Pfkelch13* gene has increased in geographic scale, frequency and local capacity [4,5,17]. The value and informativeness of ART-R molecular surveillance depend on prompt analyses and data sharing [18]. Molecular surveillance involves monitoring defined mutations within and around the propeller domain of the *Kelch 13* gene in *P. falciparum* (*Pfkelch13* markers), as these are associated with ART-R in clinical and laboratory settings. Consequently, these mutations serve as important genetic markers for the surveillance of the parasite population in infected patients [9].

East African countries, including Rwanda [15,19–21], Uganda [3,22–26], South Sudan [6,27], Tanzania [24,28], Ethiopia [29,30], Eritrea [31,32], Kenya [24], and the Democratic Republic of Congo (DRC) [6,33], report an increasing prevalence of *Pfkelch13* mutations [10]. These mutations often coincided with reduced ACT treatment efficacy [4,22,28,34–38]. There are also reports of increases in *Pfkelch13* markers in the lower transmission zones of southern Africa [39–41]; however, limited data are available on ACT efficacy in this region [42]. In the face of ART-R emergence in SE-Africa, there is a growing need to collate up-to-date resistance information at regional, national, cross-border, and sub-national levels to inform tailored approaches across diverse African settings [9].

Despite recent investigations on ART-R in Africa [9,10,16,43,44], considerable knowledge gaps remain regarding the emergence and distribution of ART-R *Plasmodium* strains across the continent [1,4–6,17]. The relatively limited coverage of molecular surveillance and ACT Therapeutic Efficacy Study (TES) outcomes, with an absence of accessible information on neighboring countries' resistance profiles and treatment guidelines, hinders coordinated malaria control and elimination efforts. Outdated clinical and molecular surveillance data further compound the barriers posed by this paucity of data [8]. The high mobility of populations within and across African borders allows resistant malaria strains to spread rapidly across national boundaries, necessitating coordinated cross-border surveillance and interventions [26]. Disease surveillance data must be shared promptly and be easily interpretable to effect timely changes in the public health response.

To address these knowledge gaps, we have developed a dashboard that collates data at the regional level, dedicated to gathering, interpreting, and visualizing up-to-date information on ART-R and the efficacy of ACTs specific to the SE-African region. This dashboard, developed by the Mitigating Antimalarial Resistance Consortium for South and East Africa (MARC SE-Africa) at the request of national malaria programs and their partners, was created to support national malaria programs and their partners. This user-friendly dashboard supports research and policy development and facilitates a coordinated response to drug-resistant malaria at national, sub-national, and regional levels. Here, we outline the dashboard's development, features, and potential impact on resistance management in support of malaria control and elimination efforts in SE and sub-Saharan Africa.

## Results

### Dashboard format overview

The landing page of the MARC SE-Africa dashboard, accessible at https://www.samrc.ac.za/antimalarial-drug-resistance, presents interactive data visualizations on antimalarial resistance. This includes molecular surveillance of *Pfkelch13* genetic markers, national malaria treatment profiles, and statistics on cases and deaths (Fig 1). Users can navigate

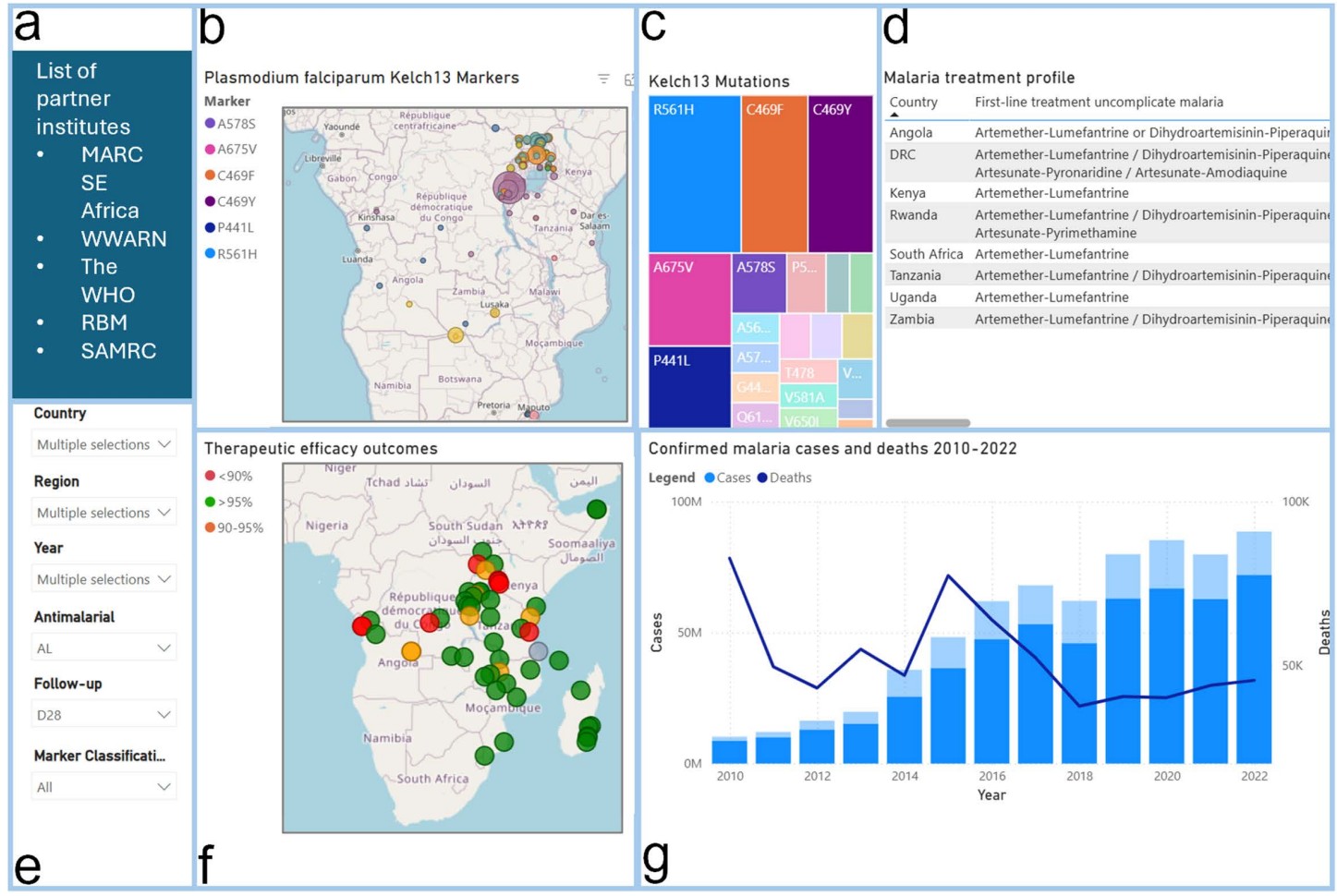

**Fig 1. Screenshot illustrating the layout of MARC SE-Africa Dashboard[1,2].** a) links to external resources complementing MARC SE-Africa data; b) map illustrating the spatiotemporal changes in *Pfkelch13* prevalences; c) treemap indicating the *Pfkelch13* markers' relative proportions; d) scrollable table displaying a malaria treatment profile per country; e) filterable parameters to customize the displayed data; f) the therapeutic efficacy study outcomes; and g) the combined graph illustrates the changes in malaria cases and deaths annually across SE-Africa (2010-2022). [1]Blue blocks added to facilitate discussion [2]The Basemap data for panels Fig 1b and Fig 1f were generated using OpenStreetMap.

through interactive maps, charts, and tables, utilizing filters such as "Country," "Region," "Year," "Antimalarial," and "Marker classification" to tailor visualizations to specific analytical needs.

The dashboard displays the geographic distribution and prevalence of WHO-validated and -candidate *Pfkelch13* genotypes associated with ART-R, alongside a color-coordinated tree map showing proportions of *Pfkelch13* markers. Additionally, the TES outcomes are illustrated on a map with color-coded markers according to WHO thresholds. Green represents TES outcomes with efficacy above 95%, orange represents outcomes between 90% and 95%, and red indicates efficacy outcomes below 90%.

### Insights from collated data on ART-R

**The *Pfkelch13* marker prevalence in eastern and southern Africa.** Our curated *Pfkelch13* database included 462 entries for 15 countries extracted from 64 sources (S1 Text). From 2014 to 2024, 73,364 samples were collected and

subjected to *Pfkelch13* sequencing and bioinformatic analysis in SE Africa, with Uganda and Rwanda having the most entries and the broadest geographical surveillance coverage. Although 23 different *Pfkelch13* markers were detected in SE-Africa, most parasite isolates (91.6%) analyzed carried the wild-type (artemisinin susceptible) phenotype.

The most prevalent validated and candidate markers detected were R561H, C469F, C469Y, A675V, and P441L (Fig 2). In contrast, the WHO-validated markers F446I, N458Y, M476I, Y493H, I543T, and P553L and the WHO-candidate markers A481V, P527H, N537I/D, G538V, and V568G were not detected in any of the studies assessed. The R561H mutation was identified in 54.5% of samples from Kibingo, Rwanda, in 2022 [45], the A675V mutation in 30% (9/30) of samples collected in Kapchorwa, Uganda, 2022 [25], and the C469F marker was detected in 40.3% of samples from Rukiga, Uganda, 2021. The non-validated A578S marker was the sixth most prevalent *Pfkelch13* mutation in SE-Africa, reaching a prevalence of up to 21.8% (S1 Table). It was detected in South Africa near the Mozambican border, Zambia, Angola, northern and central Uganda, including near the Kenyan border, Kenya [46], and Northern and Central Uganda (Fig 2).

Madagascar (2018), Burundi (2023), Botswana (2012–2014), Comoros (2017), and Malawi (2014, 2016) reported only wild-type *Pfkelch13* genotypes, while Somalia (2016–2018) had a single isolate [1/138] harboring the R622I marker in 2016. Countries without *Pfkelch13* prevalence data included Botswana, Eswatini, and Zimbabwe in southern Africa, and South Sudan and Zanzibar (Tanzania) in East Africa.

**Cross-border prevalence of *Pfkelch13*.** The R561H marker was frequently reported in cross-border regions of East African countries (Fig 2). The R561H mutation was first reported from the Southern border of Masaka, Rwanda [47], and neighboring Burundi and Tanzania in 2014. In the same year, this marker was detected to have a low prevalence in Kinshasa, DRC (0.08%). By 2015, the prevalence of R561H had increased to 7.53% in Masaka [48]. In 2017, 1.43% of *P. falciparum* samples from Buhigwe, located at the northwestern border of Tanzania, harbored the R561H marker [49]. The following year (2018), R561H was detected at a high prevalence (19.51%) on the Eastern border of Rwanda (Rukara) [15], neighboring Tanzania. In Kenya (2021), R561H was reported (7.69%) close to the southwestern border of Kenya (Kissi Country) [50], neighboring Tanzanian, which is also near the Ugandan border (~50km). In 2021, R561H was prevalent in the Kagera region (7.7%) [51], which borders Uganda, Rwanda, and near Burundi. Similarly, R561 was detected (22.54%) in Karagwe, Tanzania (2022) [26], and was also prevalent in the neighboring Kanungu (2%) [25] and Rukiga (22.6%) [25], Uganda, during the same time (2022). The R561H *Pfkelch13* marker was frequently reported in towns and cities adjacent to or in proximity to Lake Victoria, including Chato [49], Kagera [51], Karagwe District in Tanzania [28], Kisii County in Kenya [50], and Jinja in Uganda [25].

The C469F *Pfkelch13* marker was prevalent in cross-border regions in Rwanda and Uganda (Fig 2). C469F was first detected in Uganda in 2016 in two locations: Rukiga (19%), bordering Rwanda, and Tororo (2.9%), bordering Kenya [25]. In Rukiga, the C469F marker increased in notable prevalence in 2021 (40.3%) and 2022 (38.3%) [25]. The C469F marker was also reported in other cross-border regions in Uganda, including Hoima (neighboring DRC; 2018, 2.4%), Kaabong (neighboring Kenya; 2021, 5.3%), and Kanungu (neighboring DRC; 2018, 2.8%; 2021, 4.3%) [25]. Additionally, the C469F marker was reported in Bugarama near the Burundi border in southern Rwanda (2018; 1,18%) [15].

The *Pfkelch13* marker A675V was prevalent in East African countries during 2016–2023, including Uganda, Rwanda (Huye, 2019, 1.52%; 2021; Nyamagabe, 2022, 8.8%), Tanzania (Kagera, 2021, 0.15%; Tabora, 2021, 0.22%), and Kenya (Kakamega, 2018, 2.38%), from 2016 to 2023 [3,22,23,25,26,45,51–56] (Fig 2). The A675V marker was most prevalent in Uganda, particularly in cross-border regions of the country. These included towns and cities such as Lamwo (2018–2022; 2.8%-22.7%), Adjumani (2022; 2.5%), both sites close to the South Sudan border in Northern Uganda, and Arua (2022–2023; 2.6%-5.8%) and Koboko (2022; 12.8%) near the DRC border in Northwest Uganda; in Hoima (2022; 22.2%) and Kasese (2022; 3.3%) in Western parts of the country that is near the DRC border, and Kyangwali (2022; 4.4%); Southwestern Uganda in Kabale (2022; 4.4%) and Rukiga (2022; 5.1%) near the Rwanda Border, and Kanungu (2022; 3.1%) close to the DRC border; Eastern Uganda in Tororo (2022; 15.8%), Busia (2022; 2.3%), Kapchorwa (2022; 30%), and Jinja (2022; 1%) neighboring Kenya; and the Northwestern Uganda sites such as Kaabong (2022; 5.9%) and Karamoja (2021–2023; 8.3%) near the Kenyan border.

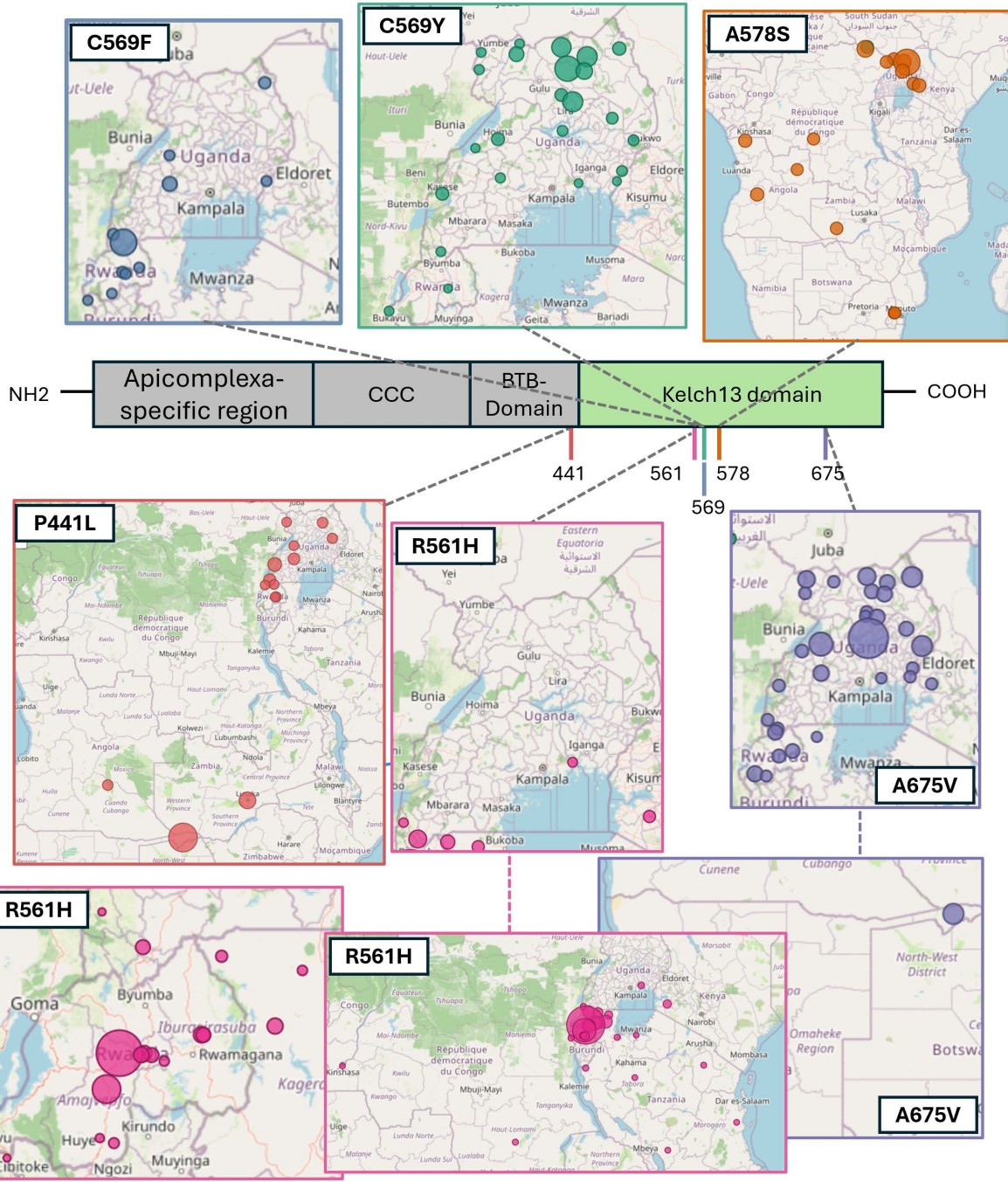

**Fig 2. Structure of the *Pfkelch13* protein, indicating the *Kelch13* domain in green and the amino acid position of genetic markers used for genomic surveillance[1].** Each marker's geographic distribution (2014-2023) and corresponding amino acid positions in the *Pfkelch13* protein are indicated. [1]The Basemap data were generated using OpenStreetMap.

Compared to eastern Africa, less surveillance was conducted in southern Africa. Recent surveillance efforts (2023) report high prevalences (38.1%) of the WHO-candidate marker P441L, and the WHO-validated markers A675V (1.99%) and P574L (1.99%) in the Zambezi region of Namibia, located between the borders of Angola, Zambia, Botswana, and Namibia [42].

**Therapeutic efficacy outcomes.** Our database currently includes 118 entries from 50 sentinel sites across 16 countries in SE-Africa from 33 references documenting TES outcomes for five WHO-recommended antimalarials: Artemether-Lumefantrine (AL), Artesunate-Pyronaridine (ASPY), Dihydroartemisinin-Piperaquine (DP), and Artesunate-Amodiaquine (ASAQ). Four southern African countries, *i.e.,* Botswana, Eswatini, Namibia, and Zimbabwe, have not yet reported any ACT TES outcomes. Five countries, Angola, 2019 and 2021; the DRC 9 (2017); Kenya, (2016); Tanzania, (2022); and Uganda (2018, 2022), reported the AL Day 28 PCR-corrected ACPR rates below the WHO-recommended threshold of 90% in their most recent TES (Fig 3A). The efficacy of ASAQ remained above 95% in all countries investigated (Angola, DRC, Madagascar, Mozambique, Tanzania, Uganda, Zambia, and Zanzibar), bar Zaire in Angola (2019), and Rutshuru in the DRC (2018) (Fig 3B). For DP, the Day 28 PCR-corrected ACPR rates of Angola (2021), DRC (2018), Kenya (2017), Malawi (2021), Somalia (2017), Uganda (2023), Zambia (2022) remained above 95% for all sites investigated. The Day 28 efficacy of PA ranged from 93.6% in Arua, Uganda (2023), to 98.1% in Comoros, Grande Comore Island (2019).

**Malaria treatment profiles in East and Southern African countries.** Artemether-lumefantrine is most widely employed as the first-line treatment for uncomplicated malaria. It is the only ACT available in three countries (Botswana, Comoros, and South Africa), and AL is used in combination with DP in Angola (DP or Injectable Artesunate-Amodiaquine), the DRC, Rwanda, Tanzania, Zambia, and Zimbabwe. Moreover, AL is employed in combination with ASAQ in Mozambique, Namibia and Zanzibar. In other countries, ASAQ is the only first-line treatment available (Madagascar, South Sudan, and Zanzibar). Multiple first-line treatments were available in Rwanda and the DRC [1]. Injectable artesunate is the primary first-line treatment for severe malaria in most countries, except for Comoros and Namibia, which still use injectable quinine despite its higher in-hospital mortality rates [57,58]. Second-line treatments for severe malaria include injectable quinine (Angola, Malawi, and Zimbabwe) or injectable artemether (the DRC, Eswatini, Kenya, Rwanda, South Sudan, Tanzania, Uganda, and Zanzibar). Pre-referral treatment with rectal (or intramuscular) artesunate was available in Angola, Botswana, Burundi, the DRC, Madagascar, Malawi, Tanzania, Uganda, Zambia, and Zimbabwe. Daily primaquine (PQ) for 7–14 days is recommended for radical cure of *P. vivax* and *P. ovale* in Botswana, Eswatini, Kenya, Madagascar, Malawi, Rwanda, South Africa, South Sudan, Zanzibar, and Zimbabwe, while single low-dose PQ is used for transmission-blocking of *P. falciparum* in Botswana, Burundi, Eswatini, Namibia, South Africa, and Somalia [1].

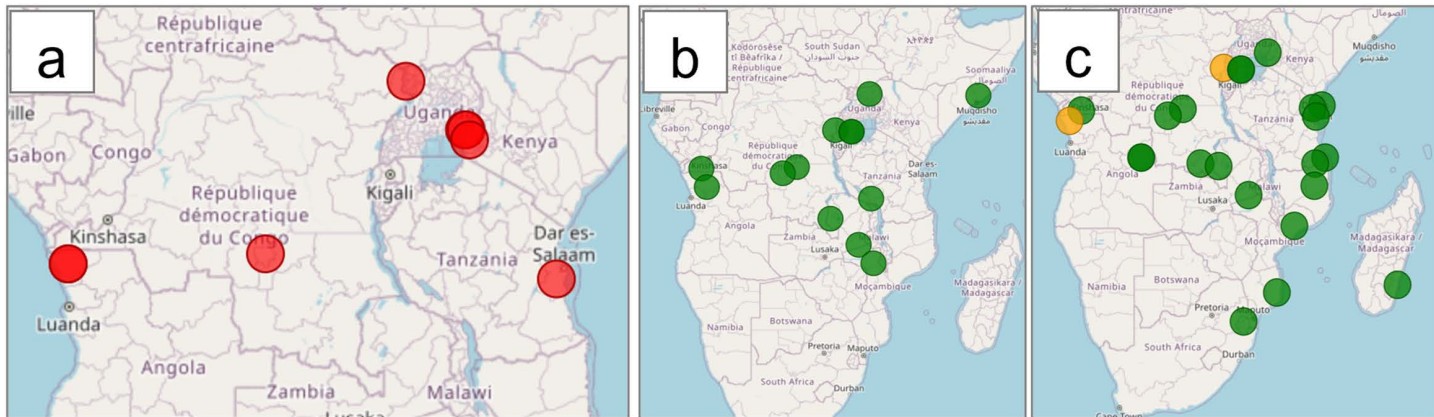

**Fig 3. Maps[1] illustrating the most recent PCR-corrected therapeutic efficacy study outcomes in SE-African countries (Angola, 2021; the Democratic Republic of Congo, 2017; Tanzania, 2023; Kenya, 2016; Zambia, 2017; and Somalia, 2016).** a) Geographic distribution of Day 28 Artemether-lumefantrine therapeutic efficacy study results below 90%, shown in red; b) Geographic distribution of Day 28 Artesunate-Amodiaquine therapeutic efficacy study results, efficacies above 95% are shown in green, and efficacies between 90-95% are shown in orange; c) Geographic distribution of Day 28 DP therapeutic efficacy study results. [1]The Basemap data were generated using OpenStreetMap.

The prevalence of *P. falciparum* mono-infection ranges from 71% in Zanzibar to 100% in most other countries. Zanzibar reported a 0.2% prevalence of *P. ovale* and 28.8% of co-infections with *P. falciparum* and *P. malariae*.

**Malaria cases and deaths reported in Southern and Eastern Africa.** Our database captured 248 entries by country and year, and the dashboard displays the total reported cases and deaths since 2010 in the 19 endemic southern and eastern African countries (including Zanzibar as a separate territory; Fig 1). In 2022, ~55,000 deaths and >101 million malaria cases were reported from these 19 countries, with 16,594 deaths and 34,448,708 cases from the southern African countries, and 21,267 deaths and 61,764,197 cases from the eastern African countries.

Malaria transmission intensity varied across the SE African regions. The DRC, Uganda, Tanzania and Mozambique accounted for a substantial proportion of the region's malaria cases (60.11%; 57,833,663; 2022) and deaths (38.53%; 14,588; 2022). The DRC, in East-Central Africa, had the highest absolute and population-adjusted death rates, notably a sharp rise in 2019 (92,232 and 113.82 per 100,000 individuals, respectively). The results show the DRC's high malaria mortality both in absolute and population-adjusted numbers, with Angola and South Sudan (2022; 40 per 100,000 and 4,429 absolute deaths) also experiencing substantial mortality rates relative to their population sizes. Between 2010 and 2022, the DRC and Tanzania reported the highest number of malaria cases (S2 Table). From 2010 to 2013, Tanzania recorded the highest number of cases, and the DRC consistently recorded the most cases from 2014 onward. Population-adjusted malaria cases, however, were highest in Burundi, with peaks in 2016 (79,593 per 100,000) and 2019 (90,151 per 100,000), when cases per 100,000 reached over 90,000.

The pre-elimination countries, Eswatini, Botswana, Somalia, Zanzibar, and South Africa, had the lowest reported cases and deaths, accounting for 0.01% (13,807) of all cases and 0.11% [44] of all deaths reported in SE-Africa in 2022. Zanzibar, a semi-autonomous region of Tanzania, reported 4,557 malaria cases in 2022, considerably lower than mainland Tanzania (3,656,651). For malaria-related mortalities, the Southern African country Malawi recorded higher population-adjusted mortality rates in 2011–2012, 45 per 100,000 and 37 per 100,000, respectively, while Angola showed decreasing rates in the later years (2019: 66 per 100,000, 2020: 35 per 100,000; and 2022: 35 per 100,000; S2 Table).

## Data accessibility

To complement the data presented (S3 Table), links to partner institutes were provided for further details and investigation. The dashboard provides quick links to the latest WHO World Malaria Report [1], the Roll Back Malaria (RBM) dashboard, downloadable, country-specific reports compiled by the President's Malaria Initiative (PMI) and MARC SE-Africa, and the WorldWide Antimalarial Resistance Network (WWARN) tools and resources page. The focus on *Pfkelch13*, therefore, provides a detailed view of ART-R, while links to external tools such as the WWARN Molecular Surveyor and WHO Malaria Threats Maps allow users to examine partner drug and diagnostic resistance markers in parallel, so that the dashboard complements rather than replaces broader molecular surveillance resources.

To ensure the broadest possible reach and impact, the dashboard is accessible via an open-access web-based system. It is maintained by the South African Medical Research Council (SAMRC), is available at https://www.samrc.ac.za/anti-malarial-drug-resistance, and can also be accessed via the MARC SE Africa web page at https://www.marcse-africa.org/antimalarial-resistance-dashboard. It works with all major web browsers, including Mozilla Firefox, Google Chrome, Safari, and Microsoft Edge. The MARC SE-Africa dashboard is free and does not require login details.

## Dashboard use cases

**Scenario 1.** In a hypothetical scenario, an NMCP manager needs to update health officials on the current status of ART-R in Tanzania. Using the dashboard "Country" parameter, the NMCP manager selects Tanzania as well as neighboring countries, namely Kenya, Uganda, Rwanda, Burundi, the DRC, Zambia, Malawi, and Mozambique, for further insight into the emergence of resistance in Tanzanian cross-border regions. An overview of the most recent TES results per site, the prevalence of *Pfkelch13* mutations, current malaria treatment policies, and annual reported cases and deaths per country was displayed (Fig 3).

The effectiveness of AL (Fig 4A) in Tanzania is above 95% in Kagera (2022) and Mbeya (2023), between 90–95% in Kigoma and Tanga, and below 90% in Pwani. All Tanzanian sites reported 100% PCR-corrected efficacy for ASAQ (Fig 4B) [28,59,60]. Fig 4C displays the latest AL efficacies in neighboring and cross-border regions. Ugandan and Kenyan cross-border sites have efficacies below the WHO-recommended 90% threshold, while Burundi reported an efficacy of 90–95%.

In Tanzania and its neighbors, AL is the first-line treatment for uncomplicated falciparum malaria, with DP as an alternative (DP, however, is exclusively available in the private sector) (Table 1). The predominant parasite species in Tanzania was *P. falciparum*, accounting for nearly all reported malaria infections. Malaria cases decreased from ~13 million in 2010 to ~3.7 million in 2022. However, malaria cases increased between 2015 and 2022 (Fig 5). Malaria deaths in the country have declined since 2010, but smaller increases were detected in 2015 and 2020 (Fig 5).

Tanzania's most prevalent *Pfkelch13* markers (Fig 6) were the WHO-validated R561H, A675V, and R622I mutations [28,45,49,51,61]. The R561H marker was more prevalent in the northwest region of Kagera, bordering Rwanda and Uganda, with the highest frequency of 35.2% reported in Rwenkende on the northwestern border of Tanzania in 2023 [45]. The R561H has been reported in six other regions in Tanzania, where the R6221I and A657V-validated ART-R markers have also been reported [28,45,49,51,61]. Of the 8,460 samples investigated (2017–2023), 8,398 (99.2%) had the wild-type *Pfkelch13* genotype.

By utilizing the MARC-SE Africa dashboard, the NMCP manager gained a comprehensive overview of ART-R status and ACT efficacy nationally and regionally, including in cross-border areas. These results were contextualized in relation to malaria treatment policies and changes in malaria morbidity and mortality. This enhanced surveillance capability supported research findings, facilitating data-driven decision-making and informed policy application across various levels. These data support the Tanzanian Ministry of Health's recent decision to replace AL with ASAQ as first-line treatment for uncomplicated malaria, with phased implementation starting in the northwestern region where *Pfkelch13* mutations are most prevalent.

**Scenario 2.** A researcher was investigating the correlation between *Pfkelch13* markers and delayed parasite clearance in the Adjumani district of the West Nile region in northwest Uganda [22]. Delayed parasite clearance, observed in 24 of the 100 enrolled patients following treatment with AL, was statistically associated with the C469Y marker ($p = 0.0278$;

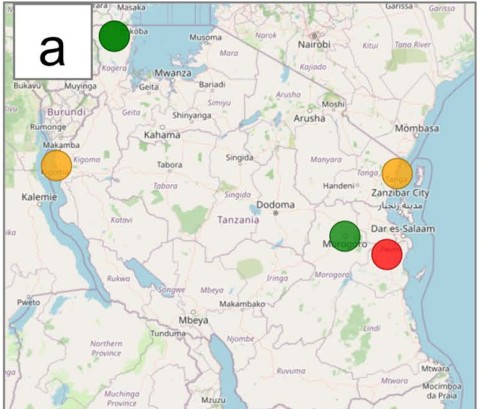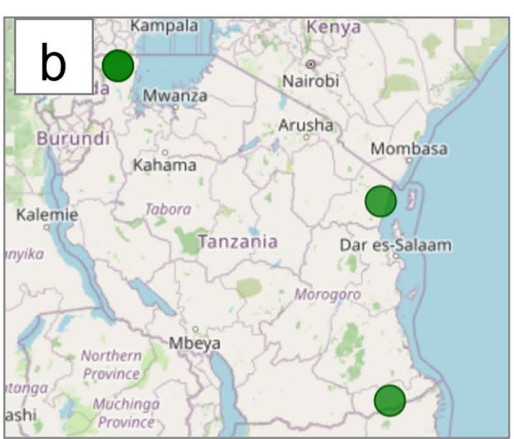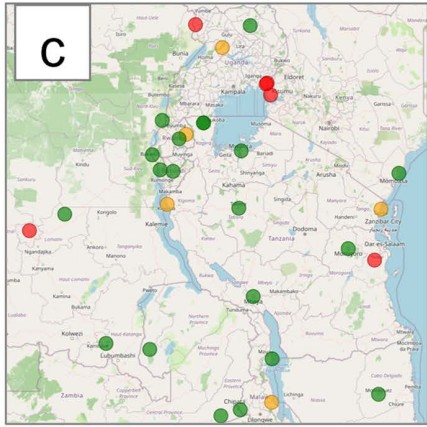

**Fig 4. Map[1] illustrating Tanzania's most recent (2023) therapeutic efficacy study outcomes 2023.** A) Geographic distribution of Day 28 Artemether-lumefantrine therapeutic efficacies above 95% are shown in green, between 90-95 in orange, and below 90%, shown in red B) Geographic distribution of Day 28 Artesunate-Amodiaquine therapeutic efficacies, efficacies (Tanzania, 2023; Uganda, 2023 and 2019; Kenya, 2016 and 2018; Rwanda, 2018) C) Geographic distribution of Day 28 Artemether-lumefantrine therapeutic efficacies of Tanzania and neighbouring countries (Tanzania, 2023; Uganda, 2023 and 2019; Kenya, 2016 and 2018; Rwanda, 2018). [1]The Basemap data were generated using OpenStreetMap.

**Table 1. Outline of the current national treatment guidelines of Tanzania.**

| Treatment criteria | Description |
|---|---|
| First-line of treatment for uncomplicated malaria | Artemether-Lumefantrine or Dihydroartemisinin-Piperaquine |
| Second-line treatment of uncomplicated malaria | Not available |
| *Plasmodium ovale* and *Plasmodium vivax* Radical cure | Not available |
| Severe malaria first-line treatment | Injectable Artesunate |
| Severe malaria second-line treatment | Injectable Artemether |
| *P. falciparum* transmission-blocking | Not in policy |
| Pre-referral treatment | Intramuscular/ Rectal Artesunate |
| Percentage *Plasmodium falciparum* (%) | 100 |

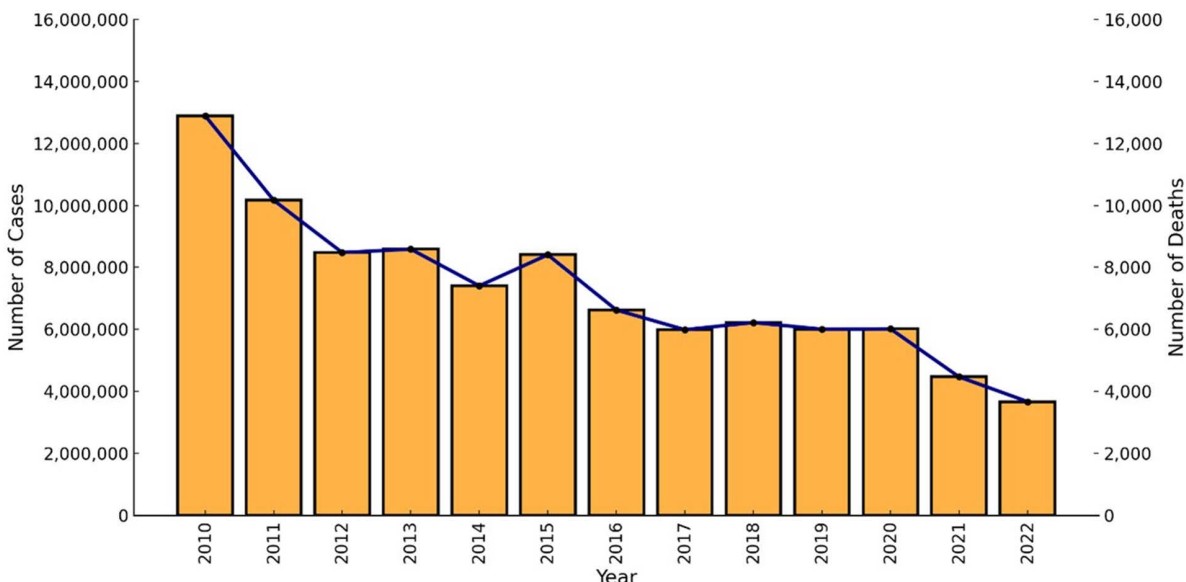

**Fig 5. Trends and changes in malaria cases (y-axis) and deaths (alternative y-axis) between 2010 and 2022 in Tanzania.**

Hodges-Lehmann estimation 3.2108 on the log scale; 95%CI 1.7076, 4.4730) [22]. By filtering for data on the C469Y marker in Uganda, the researcher identified other geographical regions where the C469Y marker was reported, its prevalence in Uganda, and relevant references to review [3,22,23,25,26,45,55,56,62].

From the data extracted using the dashboard, several valuable insights were gained.

- While the C469Y marker was widely distributed across Uganda, it was more prevalent in communities bordering or adjacent to the national boundaries of Uganda (Fig 2), such as Adjumani, Lamwo, Kitgum, and Koboko bordering South Sudan; Arua, Kasese, West Nile region bordering the DRC; Rukiga bordering Rwanda; Busia and Tororo bordering Kenya; and Kaabong and Koboko located close to the DRC and Kenyan borders.

- The C469Y marker was first detected in a single isolate in 2014, increasing to 180 isolates from numerous sites (Adjumani, Agago, Arua, Busia, Kaabong, Kapchorwa, Kasese, Katakwi, Koboko, Kole, Kyangwali, Lamwo, Rukiga, Tororo,

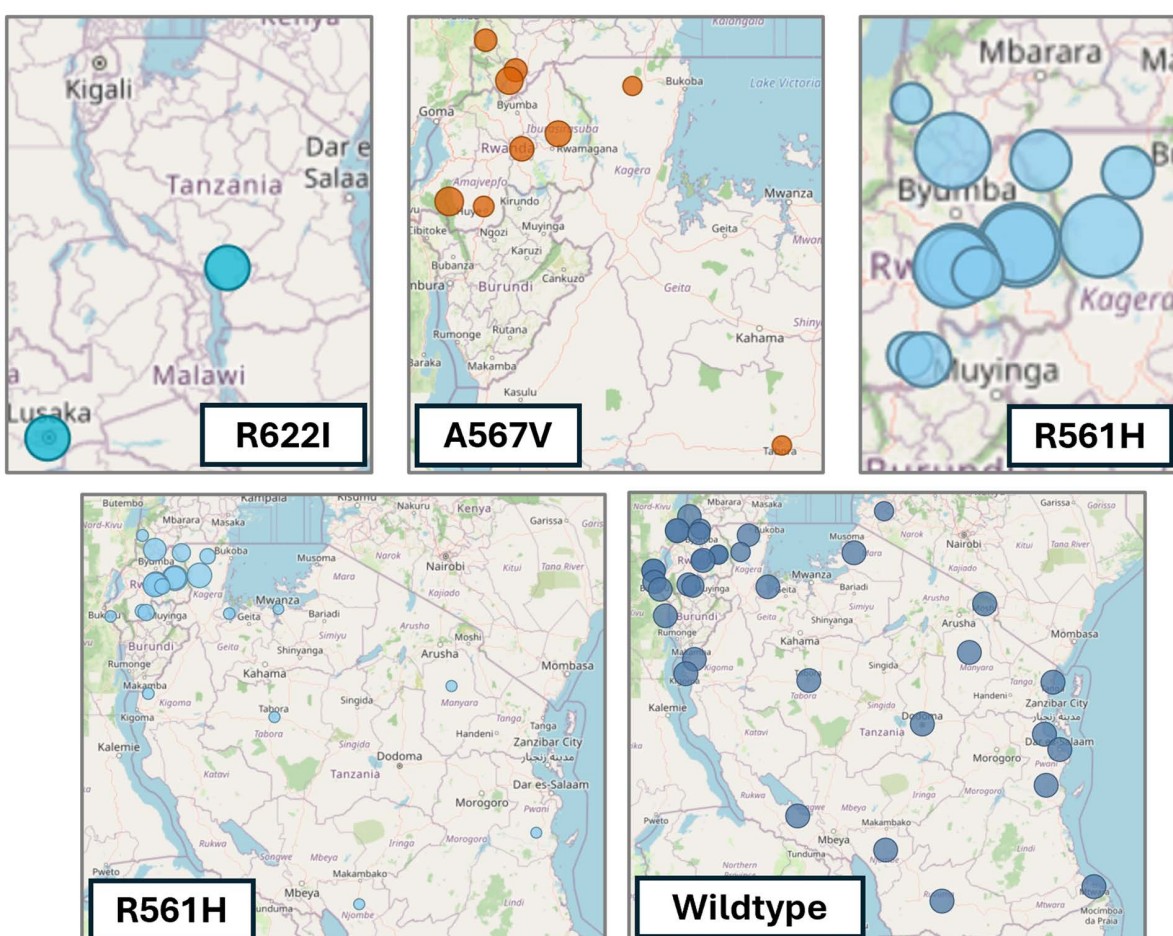

**Fig 6. The distribution¹ of *PfKelch 13* markers in Tanzania and neighboring countries.** R622I is illustrated in light blue, A675V in orange, and R561H in blue; the samples with wild-type *PfKelch13* are shown in dark blue (2017-2022). ¹ The Basemap data were generated using OpenStreetMap.

and Acholi) by 2022. The overall prevalence (total number of samples that harbored the C469Y marker) in Uganda remained high from 2016 (7.14%) to 2023 (9.6%). The highest prevalence was observed in 2020 (17.20%) (Fig 7).

- The dashboard-generated map illustrated the recent TES outcomes for Day 28 PCR-corrected cure (adequate clinical and parasitological response, ACPR) rates following treatment with AL (based on preliminary reports). The researcher finds that sentinel sites experiencing treatment failures exceeding the 10% WHO threshold also frequently reported the C469Y marker (Fig 2 and Fig 4) [35,56,63].

- The use of the MARC SE-Africa dataset and dashboard further showed that the C459Y was only reported from a single sample originating from neighboring Northern Rwanda (2014; 2% of samples) [64] and a single isolate from Bukavu in the DRC (2022; 2% of samples) [33]. Therefore, Uganda's high prevalence and wide geographic spread of the C469Y marker were localized, and findings were not shared amongst other East and southern African countries. The investigation of the MARC SE-Africa reference list of the *Pfkelch13* dataset led the researcher to recent findings that reported the independent and recent emergence of this allele originating from within Uganda [52]. From this MARC SE-Africa dashboards list of curated literature, the researcher finds that the C469Y marker most likely experienced positive selection

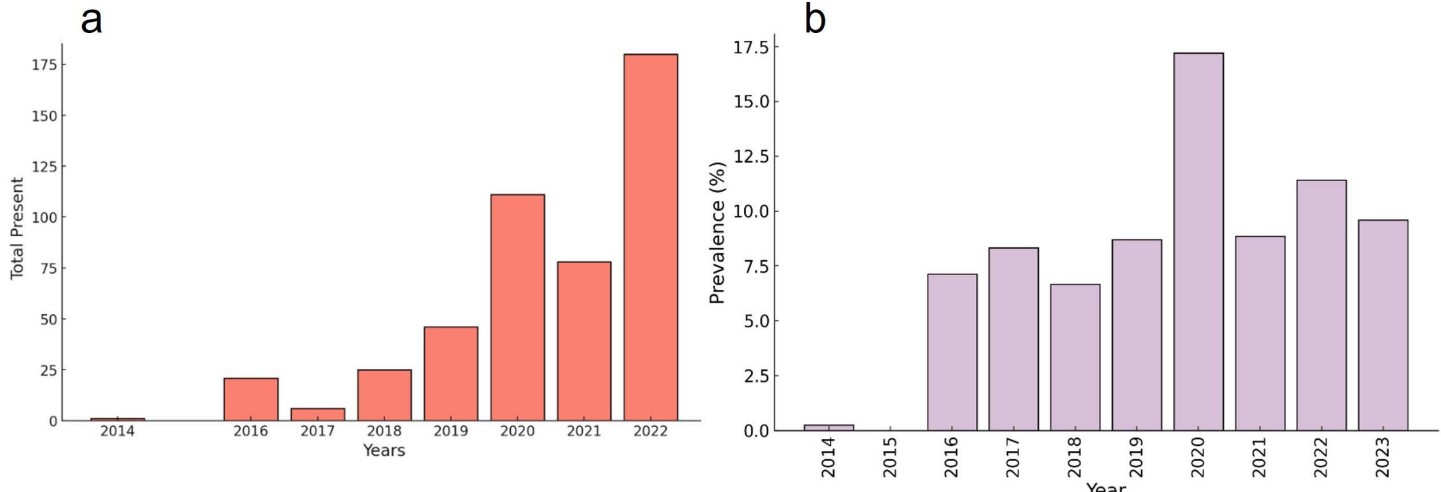

**Fig 7. a) The total number of isolates harboring the WHO-validated C469Y marker of ART-R resistance sampled in Uganda, 2014-2022.** b) The prevalence of the C469Y marker in Uganda (2014-2022).

within Ugandan *Plasmodium* populations, which supports the observation of its sustained high prevalence and geographic distribution [52].

- Using the graphs and maps generated with the MARC SE-Africa dashboard, the researcher could further support their discussions on the potential risk of cross-border movement of parasites with the C469Y marker, particularly in areas with highly mobile populations, as seems apparent for other WHO-validated markers of ART-R, such as R561H or A675V (Fig 2).

The data and references sourced from our dashboard could bolster the researcher's findings and contextualize their work with those published previously by exporting these figures and linking the findings to the relevant studies available from our curated reference list.

## Discussion

The MARC SE-Africa dashboard was designed at the request of NMCPs to support a detailed analysis of ART-R in SE-African countries, which carry among the highest global malaria burdens and where ART-R and ACT failures are increasingly reported [1]. Our dashboard provides information on antimalarial treatment modalities included in national malaria treatment policies and their efficacy; annual malaria morbidity and mortality data since 2014; and trends, prevalence, and spatiotemporal distribution of ART-R molecular markers. This user-friendly and open-access web-based tool facilitates informed decision-making and efficient targeting of appropriate interventions. The dashboard was built using open-source software to ensure its sustainability and ease of maintenance. Its functionality was iteratively refined based on stakeholder feedback to ensure accuracy, relevance, and utility.

Dashboards have emerged as essential tools for managing infectious diseases globally, including in Sub-Saharan Africa, as exemplified by their increased use during the COVID-19 pandemic to disseminate information, provide epidemiological statistics, and guide public health interventions [65–67]. Dashboards serve as digital tools for diverse audiences and play an essential role in directing the public health response and supporting decision-making among policymakers, public health officials, and scientists [65–69]. Our dashboard simplifies complex genomic and clinical data, making it more accessible to all stakeholders. It leverages these capabilities to offer regional insights into drug-resistant malaria, shaping

policies and steering public health strategies explicitly tailored to country- and cross-border-specific needs. Indeed, the dashboard has already informed regional planning processes: its collated datasets and country profiles contributed to the East African Community Regional Detailed Action Plan for Responding to Antimalarial Drug Resistance, endorsed by the region's ministers of health in 2025, and to a draft regional antimalarial resistance profile and action pathway for the Southern African Development Community [70–72]. In addition, partners such as Maisha Meds have drawn on the dashboard outputs in analyses and webinars that explore coordination challenges between pharmaceutical, regulatory and other antimicrobial resistance stakeholders [73].

We investigated collated data on TES outcomes in SE-Africa. The data on AL efficacy revealed a concerning trend of decreasing AL efficacy in four East African countries, which were below the WHO's recommended threshold of 90% (DRC, Kenya, Rwanda and Uganda) [1]. Less is known about the therapeutic efficacy of AL in Southern African countries, where AL efficacy has been repeatedly reported below 90% in Angola, and no efficacy data are available for Eswatini, Namibia, and Botswana. Encouragingly, the efficacies of DP and ASAQ in SE-Africa remain above the WHO-recommended efficacy threshold, suggesting that they could replace AL. However, extensive DP use can accelerate the selection of molecular markers associated with resistance to artemisinin derivatives and piperaquine [74]. This could increase treatment failures, especially when DP is used concurrently with treatment protocols and chemoprevention strategies. Similarly, in regions where SP and AQ are used together for seasonal malaria chemoprevention, ASAQ may not be considered an ideal first-line treatment. Moreover, *Pfcrt* mutations associated with AQ resistance were reported in the DRC and along the southern Rwanda-Burundi border [76], where ASAQ was heavily utilized until 2020 [1,76,77]. Multi-level and multi-national evidence is therefore needed to allow sub-national and cross-border insights to combat malaria in the face of ART-R.

Given the high mobility of populations across Africa and the cross-border prevalence of ART-R markers, comprehensive surveillance and regional collaboration are essential for informed and effective public health responses [26,75]. We report that the C469F/Y, R561H, P441L, A578S, and A675V markers had high cross-border prevalences, particularly in East Africa. Indeed, a recent investigation into the prevalence of ART-R among refugees arriving in Uganda from South Sudan and the DRC reported high prevalences of the validated markers A675V and C469Y [26]. Similarly, the dashboard collated data revealed concentrations of ART-R markers and AL treatment failures in countries surrounding the Great Lakes of East Africa. Two cities, Jinja in Uganda and Bukoba in Tanzania's Kagera region, have established harbours on Lake Victoria, which may facilitate cross-border transmission and, in turn, the spread of the R561H allele or others throughout the region. Moreover, although surveillance from most southern African countries has yet to report high levels of ART-R markers, the movement of asymptomatic malaria refugees from eastern and central regions of Africa may facilitate the spread of ART-R into low-endemic regions [75]. Taken together, continued surveillance of cross-border regions and mobile populations will be required to monitor the spread of ART-R and enable informed public health responses.

An in-depth investigation of the references and datasets curated for our dashboard enabled the identification of spatiotemporal trends in the spread and emergence of ART-R in East Africa, including the current status of ART-R in Tanzania and the distribution and prevalence of the WHO-validated C469Y marker in Uganda. In this study, we described the clinical, genetic, and evolutionary significance of C469Y in Uganda [22,52]. Discussions facilitated by examining our curated literature found that the high prevalence and wide geographic spread of the C469Y in Uganda was unique among SE-African countries. However, similar distribution and high prevalence have been reported previously in Southern and Eastern Asia [76]. The C469Y marker is statistically significantly associated with prolonged parasite clearance half-lives under experimental conditions [76] and parasitemia that persists on Day 2 or 3 after AL treatment during clinical trials [22]. Combatting malaria in SE-Africa requires an in-depth understanding of both the clinical and genetic landscape of ART-R.

Three validated *Pfkelch13* markers, A675V, R561H, P574L, and the candidate marker, C469F, were prevalent in cross-border regions of Tanzania, Uganda, and Rwanda. The high prevalence of the non-validated A578S marker throughout SE-Africa is of some concern. While its clinical significance and influence on treatment outcomes remain unknown [2,22,52], its sustained prevalence in the region suggests that this allele may confer a selective advantage

for survival and transmission. Therefore, further research is needed on the significance of this marker. The absence of specific validated markers, such as the WHO-validated markers F446I, N458Y, M476I, Y493H, I543T, and P553L and WHO-candidate markers A481V, P527H, N537I/D, G538V, and V568G, was not detected in any of the studies assessed, which suggests genetic heterogeneity between the *Plasmodium* populations in Asia and Africa [10,16]. These regional and functional variations illustrate the need for continued surveillance of known and novel resistance markers using next-generation sequencing technologies to identify trends and develop up-to-date guidance for treatment decisions in Africa.

The dashboard's multinational dataset has exposed evidence gaps and highlighted areas for research and surveillance. Molecular surveillance efforts in Burundi, South Sudan, Somalia, Botswana, and Eswatini need to be strengthened, as data here is either outdated or absent. Given the lower caseload, the limited data from eliminating malaria in southern Africa are understandable but remain problematic, as drug-resistant malaria frequently emerges and spreads in areas of low transmission intensity, where individuals lack partial malaria immunity and drug pressure is higher [4,5,55]. The WHO recommends that the TES of first-line treatments for uncomplicated malaria be evaluated at least every two years in areas with ongoing transmission [9]. This regular assessment helps monitor drug efficacy, detect emerging drug resistance early, and guide potential updates to treatment policies. Data on AL efficacy that were older than three years in six countries, the DRC [38], Kenya [37], Rwanda [64], Somalia [77], South Africa [78], South Sudan [79] and Zanzibar [80,81], indicating the urgent need for updated evidence [9]. These findings support the need for easily accessible, up-to-date data to provide insight beyond country-level and identify knowledge gaps.

Importantly, by displaying data at a regional scale, the dashboard also reveals where TES and molecular surveillance are absent or outdated, including in pre-elimination settings such as Eswatini and Botswana. These visible gaps can support national programs, regional bodies, and donors to priorities data collection and expedited evidence sharing, while recognizing that in low transmission settings, the small number of infections often limits the feasibility of adequately powered TES and molecular surveys.

Despite the dashboard's utility and advantages, its effectiveness is contingent on several factors. As this tool is web-based, reliable internet connectivity is required, which can be limited in some malaria-endemic regions. The platform's capacity to foster collaboration relies heavily on stakeholders' continuous data generation and sharing. Similarly, the accuracy of reported cases and deaths depended on the strength of healthcare systems and their ability to capture and report these values accurately. Indeed, inconsistent or inaccurately reported epidemiology data have become a cause of concern for many malaria-endemic countries of SE-Africa. Additionally, the *Pfkelch13* markers alone do not provide sufficient evidence on drug-resistant malaria, and other genetic markers may provide deeper insight into the emergence and spread of resistance in SE Africa. The dashboard and analyses presented here are descriptive in nature and were not designed to adjust fully for confounding or to make causal or predictive inferences, which will require separate, dedicated modelling studies building on this curated dataset. Moreover, the utility of the dashboard hinges on both ongoing investment in surveillance and research to keep the data relevant, as well as on decision-makers' readiness to use these data to inform and implement appropriate policies.

In conclusion, the MARC SE-Africa Dashboard represents an important development in the fight against ART-R, providing a valuable tool for data-driven decision-making. By integrating relevant data, the dashboard effectively empowers stakeholders, researchers, and NMCPs to implement informed, responsive actions. It facilitates regional cooperation, enabling countries to share insights, coordinate strategies, and enhance cross-border interventions effectively. The dashboard's potential to influence malaria policy, foster regional collaboration, and bridge crucial data gaps makes it an indispensable asset in the ongoing battle against drug-resistant malaria. Continued advancements and expansion of the dashboard will be crucial in maintaining treatment efficacy and driving the agenda for malaria elimination. Success in these efforts will depend heavily on sustained investments in surveillance, research, and collaborative initiatives across SE-Africa. The region can achieve sustainable malaria control by uniting efforts and leveraging digital tools such as the MARC SE-Africa Dashboard.

## Methods

### Dashboard overview

To aid in the understanding of the current ART-R and ACT efficacy status at various administrative levels in SE-Africa, the dashboard provides an overview of four information categories: TES outcomes of WHO-recommended antimalarial treatments, the prevalence of WHO-validated, candidate, and other African-prevalent molecular markers of ART-R (*P. falciparum kelch13* propeller gene region; *Pfkelch13*), malaria cases and deaths, and malaria treatment policies in 19 malaria-endemic SE-African countries. The 19 malaria-endemic countries that are part of the East African Community and the Southern African Development Community are Angola, Botswana, Burundi, Comoros, the DRC, Eswatini, Kenya, Madagascar, Malawi, Mozambique, Namibia, Rwanda, Somalia, South Africa, South Sudan, Tanzania (Mainland), Uganda, Zambia, and Zanzibar (Tanzania); Zanzibar is represented separately from mainland Tanzania due to its pre-elimination status.

### Data input

**Pfkelch13 *genotype prevalence*.** Systematic screening of both published and unpublished literature, *i.e.*, those not yet readily available in the public domain, such as conference presentations, molecular datasets, and online resources (such as the WHO Malaria Threat Maps), was mined for *Pfkelch13* ART-R molecular marker prevalence data (Table 2, S1 Text, S2 Text, S3 Text). These data were extracted and mapped to determine the spatiotemporal prevalence of *Pfkelch13* mutations in SE-African countries. The inclusion and exclusion criteria are listed in S4 Table and S5 Table. These criteria were used to screen for studies reporting on *Pfkelch13* molecular markers from samples collected between 1 January 2014 and 30 September 2024 in SE-African countries. Eligible studies included case reports, cross-sectional studies, clinical trials, observational studies, and prospective genotyping studies that reported *Pfkelch13* marker prevalence. Studies on non-falciparum species, non-human hosts, those with unknown geographic origins, or samples collected outside the defined sampling period were excluded. Table 3 lists molecular markers extracted as WHO-validated or candidate markers [9] and other markers reported to have noteworthy prevalence in SE-African countries. The clinical significance of the markers not included in the WHO list (labelled as 'Other') is unknown; however, they were included to provide a comprehensive overview of their potential role in regional resistance patterns and to identify emerging trends that may warrant further investigation.

S6 Table and S4 Text lists the data extracted and collated for each data entry of the *Pfkelch13* marker dataset. This dataset includes information on the country, district, and site where samples were collected, geographic coordinates, sampling years, and prevalence of the detected marker. The geographical coordinates of the sampling sites were determined using Google Maps and integrated into the dashboard through Geographic Information System (GIS) mapping to

**Table 2. Databases and resources searched for *Pfkelch13* genotype prevalence data.**

| Resource | Link |
| --- | --- |
| PubMed | https://pubmed.ncbi.nlm.nih.gov/ |
| WWARN Clinical Trial Library | https://www.iddo.org/wwarn/wwarn-clinical-trials-publication-library |
| WWARN Molecular Surveyor Database | https://moldm.iddo.org/ |
| WWARN Artemisinin Molecular Surveyor Dashboard | http://www.wwarn.org/molecular/surveyor/k13/index.html?t=201608031200#0 |
| References listed in MARC SE-Africa Country Profiles | https://www.marcse-africa.org/malaria-resistance-profiles |
| Studies cited on the WHO Threats Map | https://apps.who.int/malaria/maps |

**Table 3. List of molecular marker prevalence extracted for the dashboard database.**

| Validated[1] | Candidate[1] | Other[1] |
|---|---|---|
| C469Y | C469F | S521L |
| A675V | P441L | A557S |
| C580Y | R515K | A569S |
| P574L | Wildtype | A569T |
| R539T | P553L | A578P |
| R561H | A481V | A578S |
| R622I | P527H | A613S |
| G449A | N537I/D | Q613E |
| F446I | G538V | T478 |
| N458Y | | V581A |
| M476I | | V650I |
| Y493H | | V666 |
| I543T | | V568G |

[1]Genetic markers are classified according to the WHO classifications [82,83]; markers listed as "Other" are not included under WHO validated and candidate lists; however, the samples were reported to have a noteworthy prevalence from Southern and Eastern African countries.

enable accurate spatial data visualization. The complete dataset and reference list can be accessed via https://tinyurl.com/AntimalRepo.

## Outcomes of TES

Similarly, published and unpublished literature and resources were systematically mined to define TES outcomes of WHO-recommended antimalarials in the participating countries (S7 Table- S8 Table). The screened literature included prospective clinical trials involving patients with uncomplicated falciparum malaria, many of which were accessed through the WWARN Clinical Trial Publication Library (WCTL). The inclusion and exclusion criteria are listed in S7 Table. Included trials had at least one study arm in which participants received a WHO-recommended ACT and reported on clinical and parasitological responses. The primary outcomes assessed included the PCR-corrected clinical and parasitological response by Day 28 or Day 42 (S7 Table). S9 Table outlines the categories of data extracted from clinical trials investigating antimalarial efficacy in MARC SE-Africa countries. The dataset included information on the country, region (as classified according to the United Nations geoscheme), study site and year. Other variables extracted included the antimalarial drugs studied, follow-up periods, and outcomes such as PCR-corrected and uncorrected adequate clinical and parasitological responses (ACPR) and parasite positivity rates on Day 2 or 3. Geographic coordinates were determined using Google Maps and recorded for spatial mapping. The results included TES outcomes for the WHO-recommended antimalarials AL, DP, ASPY, and ASAQ.

## Malaria treatment profile by country

An overview of the malaria treatment profiles was generated based on each country's malaria treatment policies and guidelines. The data were collated from multiple sources, including the World Malaria Report [1], MARC SE-Africa country profiles (accessed via https://www.marcse-africa.org/malaria-resistance-profiles on 26/08/2024), the President's Malaria Initiative country profiles (accessed via https://www.pmi.gov/country-profiles-2024/ on 25/08/2024), and National Malaria Program Treatment Guidelines [9,84–93]. Country-specific information on the first- and second-line treatments for uncomplicated falciparum malaria, treatments for non-falciparum malaria, first- and second-line treatments for severe malaria,

pre-referral treatments, use of *P. falciparum* transmission-blocking antimalarials, and the percentage of *P. falciparum* infections reported per country was collected (Table 4).

## Yearly confirmed malaria cases and deaths per country

Data on malaria cases and deaths reported from 2010 to 2022 were extracted from the World Malaria Report and visualized on the dashboard for each country. Country partners, co-authors, and East African Community (EAC) focal point members (NMCP members tasked by the EAC; S10 Table) reviewed the World Malaria Report [1] data to confirm the accuracy, with adjustments made where appropriate. Discrepancies between NMCP and WHO data were recorded and listed in the GitHub repository for this dataset. Newly released cases and deaths will be added for each country as they become available each year. The malaria cases and deaths dataset and references are available via https://tinyurl.com/AntimalRepo. Data on each country's population were extracted from Worldometer (accessed 25/10/2024; https://www.worldometers.info/world-population) to enable population-adjusted investigation of malaria cases and deaths in SE-African countries.

## Data integration, management, availability and visualization

The data on antimalarial resistance, including TES outcomes, the *Pfkelch13* prevalence, national treatment profiles, and confirmed malaria cases and deaths, were integrated into a centralized database accessed via REDCap [94]. We implemented automated data import and validation processes in REDCap to ensure the dashboard was updated regularly as new information became available. MARC SE-Africa curates the database monthly.

The datasets were accessed via the secure REDCap application programming interface and imported into the Microsoft Power BI Query Editor. Data visualization was developed using Microsoft Power BI and published in the Microsoft Power

**Table 4. Outline and description of the malaria treatment profile per country.**

| Category | Description |
| --- | --- |
| **Country** | The MARC SE-Africa country. |
| **First-line treatment for uncomplicated malaria** | The primary antimalarial/s recommended for treating uncomplicated malaria and alternatives included in country treatment guidelines. |
| **Second-line treatment for uncomplicated malaria** | The second-line antimalarial therapy is used to treat uncomplicated malaria if the first-line treatment fails or is unavailable. |
| ***Plasmodium ovale/ Plasmodium vivax* Radical cure** | The treatment used for the radical cure of *P. ovale* and *P. vivax* infections to eliminate liver-stage parasites. |
| **Severe malaria 1st line treatment** | The recommended first-line treatment for severe malaria cases. |
| **Severe malaria 2nd line treatment** | The second-line treatment option for severe malaria cases when the first line is not effective, tolerated or available. |
| ***P. falciparum* transmission-blocking** | Interventions that prevent the malaria parasite from spreading from humans to mosquitoes through medications like primaquine, which targets the sexual stages (gametocytes). |
| **Pre-referral treatment** | The treatment is administered in the community before a patient with severe malaria is referred to a healthcare facility for further treatment. |
| **Percentage of *Plasmodium falciparum*** | The proportion of malaria cases in the country diagnosed as *P. falciparum* species. |

BI web portal. The Basemap data were generated using OpenStreetMap (https://www.openstreetmap.org/copyright). The MARC SE-Africa dashboard is embedded on a website and is a free-to-access tool available via https://www.samrc.ac.za/antimalarial-drug-resistance. All collated and generated data and Python code used for data visualization are accessible via https://tinyurl.com/AntimalRepo. The data is iteratively refined and subject to review and monthly updates by the MARC SE-Africa team.

## Supporting information

**S1 Table. Summary of the prevalence and geographic distribution of the A578S pfKelch13 genetic marker.**
(DOCX)

**S1 Text. Titles and sources of published studies used for data mining to determine the prevalence of pfKelch13 in South and East African countries.**
(DOCX)

**S2 Table. MARC SE-Africa countries reporting the highest malaria cases and deaths relative to their estimated population (per 100,000 population).**
(DOCX)

**S2 Text. Kelch13 genotyping search terms.**
(DOCX)

**S3 Table. Links to additional resources for further information on antimalarial drug resistance and resources to combat malaria.**
(DOCX)

**S3 Text. WorldWide Antimalarial Resistance Network (WWARN) Clinical Trial Publication Library details.**
(DOCX)

**S4 Table. Inclusion criteria and description of pfKelch13 genotyping results.**
(DOCX)

**S4 Text. Scoping review search terms.**
(DOCX)

**S5 Table. Exclusion criteria for literature-based searches for pfKelch13 genotyping results.**
(DOCX)

**S6 Table. PfKelch13 marker prevalence data categories extracted for each marker entry.**
(DOCX)

**S7 Table. Inclusion and exclusion criteria for screening therapeutic efficacy study outcome studies.**
(DOCX)

**S8 Table. WorldWide Antimalarial Clinical Trials Library literature search terms.**
(DOCX)

**S9 Table. Data capture variables for the therapeutic efficacy study outcome database.**
(DOCX)

**S10 Table. Country partners, National Malaria Control Programmes, and focal point members by country.**
(DOCX)

## Acknowledgments

We acknowledge the contributions of the East African Community members, National malaria control programs, The President's Malaria Initiative, The Elimination 8 Regional Malaria Molecular Surveillance Initiative, and the Mitigating Antimalarial Resistance Consortium for South and East Africa for reviewing the data and providing guidance. We thank the Medical Research Council of South Africa for hosting the dashboard on their website and assisting in its development. We further acknowledge the contributions of Marlyn Solomons for technical assistance and project management.

## Author contributions

**Conceptualization:** Dhol S. Ayuen, Donnie Mategula, Karen I. Barnes.

**Data curation:** Stephanie van Wyk, Eulambius M. Mlugu, Donnie Mategula, Tikhala Makhaza, James Kiarie, Jimmy Opigo, Kibor Kipkemoi Keitany, Pierre Sinarinzi, Peter Aguek Kon Baak, Tommy Nseka Mambulu, Samwel Lazaro Nhiga, Sijenunu Aron Mwaikambo, Maulid Kassim, Sija Joseph Sija, Abdikarin Hussein Hassan, Jaishree Raman, Karen I. Barnes.

**Formal analysis:** Stephanie van Wyk, Eulambius M. Mlugu, Tikhala Makhaza, Victor Asua, Jaishree Raman, Karen I. Barnes.

**Funding acquisition:** Karen I. Barnes.

**Investigation:** Stephanie van Wyk, Eulambius M. Mlugu, Donnie Mategula, Tommy Nseka Mambulu, Jaishree Raman, Karen I. Barnes.

**Methodology:** Stephanie van Wyk, Ishen Seocharan, Dhol S. Ayuen, Karen I. Barnes.

**Project administration:** Karen I. Barnes.

**Resources:** Ishen Seocharan, Dhol S. Ayuen, James Kiarie, Jimmy Opigo, Aimable Mbituyumuremyi, Kibor Kipkemoi Keitany, Emmah Mongina Nyandigisi, Peter Aguek Kon Baak, Maulid Kassim, Jaishree Raman, Karen I. Barnes.

**Software:** Stephanie van Wyk, Ishen Seocharan.

**Supervision:** Michael Katende, Karen I. Barnes.

**Validation:** Donnie Mategula, Emmah Mongina Nyandigisi, Michael Katende, Karen I. Barnes.

**Visualization:** Stephanie van Wyk, Ishen Seocharan.

**Writing – original draft:** Stephanie van Wyk, Jaishree Raman, Karen I. Barnes.

**Writing – review & editing:** Stephanie van Wyk, Ishen Seocharan, Eulambius M. Mlugu, Donnie Mategula, Victor Asua, Michael Katende, Jaishree Raman, Karen I. Barnes.

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
