## [Decision Letter · Decision Letter 0]

17 Nov 2025

PDIG-D-25-00012The MARC SE-Africa Dashboard: Joining Forces to Counteract Emerging Antimalarial Resistance in South and East AfricaPLOS Digital Health Dear Dr. van Wyk, Thank you for submitting your manuscript to PLOS Digital Health. We apologise that it took so long time reach a decision, but several editors and reviewers became unavailable during the process. After careful consideration, we feel that it has merit but does not fully meet PLOS Digital Health's publication criteria as it currently stands. Therefore, we invite you to submit a revised version of the manuscript that addresses the points raised during the review process. Please submit your revised manuscript by Jan 16 2026 11:59PM. If you will need more time than this to complete your revisions, please reply to this message or contact the journal office at digitalhealth@plos.org.  Please include the following items when submitting your revised manuscript:* A rebuttal letter that responds to each point raised by the editor and reviewer(s). You should upload this letter as a separate file labeled 'Response to Reviewers'. This file does not need to include responses to any formatting updates and technical items listed in the 'Journal Requirements' section below.'. This file does not need to include responses to any formatting updates and technical items listed in the 'Journal Requirements' section below.* A marked-up copy of your manuscript that highlights changes made to the original version. You should upload this as a separate file labeled 'Revised Manuscript with Track Changes'.'.* An unmarked version of your revised paper without tracked changes. You should upload this as a separate file labeled 'Manuscript'.'. If you would like to make changes to your financial disclosure, competing interests statement, or data availability statement, please make these updates within the submission form at the time of resubmission. Guidelines for resubmitting your figure files are available below the reviewer comments at the end of this letter. We look forward to receiving your revised manuscript. Kind regards, Adam Hulman, Ph.D.Section EditorPLOS Digital Health Leo Anthony CeliEditor-in-ChiefPLOS Digital Healthorcid.org/0000-0001-6712-6626 **Journal Requirements:** If the reviewer comments include a recommendation to cite specific previously published works, please review and evaluate these publications to determine whether they are relevant and should be cited. There is no requirement to cite these works unless the editor has indicated otherwise.  **Additional Editor Comments (if provided):****Reviewers' Comments:** Reviewer's Responses to Questions

**Comments to the Author**

1. Does this manuscript meet PLOS Digital Health’s publication criteria? Is the manuscript technically sound, and do the data support the conclusions? The manuscript must describe methodologically and ethically rigorous research with conclusions that are appropriately drawn based on the data presented.? Is the manuscript technically sound, and do the data support the conclusions? The manuscript must describe methodologically and ethically rigorous research with conclusions that are appropriately drawn based on the data presented.

Reviewer #1: Yes

2. Has the statistical analysis been performed appropriately and rigorously?

Reviewer #1: Yes

3. Have the authors made all data underlying the findings in their manuscript fully available (please refer to the Data Availability Statement at the start of the manuscript PDF file)?

The PLOS Data policy requires authors to make all data underlying the findings described in their manuscript fully available without restriction, with rare exception. The data should be provided as part of the manuscript or its supporting information, or deposited to a public repository. For example, in addition to summary statistics, the data points behind means, medians and variance measures should be available. If there are restrictions on publicly sharing data—e.g. participant privacy or use of data from a third party—those must be specified.requires authors to make all data underlying the findings described in their manuscript fully available without restriction, with rare exception. The data should be provided as part of the manuscript or its supporting information, or deposited to a public repository. For example, in addition to summary statistics, the data points behind means, medians and variance measures should be available. If there are restrictions on publicly sharing data—e.g. participant privacy or use of data from a third party—those must be specified.

Reviewer #1: Yes

4. Is the manuscript presented in an intelligible fashion and written in standard English?

Reviewer #1: Yes

5. Review Comments to the Author

Reviewer #1: This manuscript introduces the MARC SE-Africa Dashboard, a real-time, open-access digital tool designed to track antimalarial resistance trends in 19 countries across South and East Africa. Developed by the Mitigating Antimalarial Resistance Consortium (MARC) in partnership with national malaria programs, the dashboard integrates and visualizes data on Pfkelch13 genetic markers, therapeutic efficacy studies (TES), treatment guidelines, and malaria morbidity/mortality statistics.

By offering interactive maps and charts, the dashboard allows stakeholders to monitor resistance patterns, assess policy relevance, and identify cross-border transmission risks. The tool has already supported national-level policy changes (e.g. Tanzania’s decision to switch first-line therapies).

The manuscript is clearly written, well-structured, and uses standard academic English. Occasional long paragraphs might benefit from editorial tightening for clarity and readability, but overall the language is fluent, professional, and precise.

Please discuss or address the following limitations:

- Data Gaps: Several countries (e.g., Eswatini, Botswana) lack recent TES or molecular data.

- Limited Analytical Depth: Primarily descriptive; lacks predictivel analytics.

- Dependence on Connectivity: Dashboard requires stable internet access, which may hinder use in remote areas. Is an offline version possible?

- Surveillance Quality Variability: Relies on external data sources of varying completeness and accuracy.

- Pfkelch13 Focus: While relevant, Pfkelch13 is not the only genetic marker of resistance—dashboard may miss broader resistance trends.

Evaluation of data: The authors should discuss confounding variables. Moreover, although trends are reported, causality or prediction is not explored.

Recommendation for Authors: include literature/data and please discuss:

- Regional coordination challenges; The dashboard’s policy impact; Technical limitations of Pfkelch13-only surveillance

e.g. in detail:

1) Recent Advances in ART-R Research in Africa

2) Digital Dashboard Evaluation and Use in general

3) Cross-Border Surveillance & Migration Impact

4) Genomic Surveillance Techniques & Broader Resistance Markers (especially for Plasmodium falciparum)

Please add future directions and goals for the dashboards presented. Is the integration of predictive analytics (machine learning a possible development? What is needed to integrate it in your dashboard?).

Please discuss if expanding genomic surveillance beyond Pfkelch13 is possible?

Third, developing offline-compatible versions or lightweight mobile apps could expand access in regions with limited internet infrastructure. Is this an option? Please discuss ths use of the dashboards here.

General Observations on figures: The figures are data-rich, policy-relevant, and well-integrated into the manuscript narrative. Most figures rely on aggregated descriptive data — predictive modeling overlays could be a next step.

Methods: Why is the Method section after the discussion? Please move between introduction and results for better readability.

Finally, the MARC SE-Africa Dashboard shifts the malaria surveillance paradigm from reactive, country-level reports to real-time, data-driven, regionally coordinated public health action, specifically targeting the emerging threat of artemisinin resistance in sub-Saharan Africa.

Please discuss more how the dashboards can help/contribute to fight death/illness from malaria.

How is the situation in Africa compared to other "malaria-countrys" like Asia?

6. PLOS authors have the option to publish the peer review history of their article (what does this mean?). If published, this will include your full peer review and any attached files.). If published, this will include your full peer review and any attached files.

**Do you want your identity to be public for this peer review?** For information about this choice, including consent withdrawal, please see our Privacy Policy..

Reviewer #1: No

  **Figure resubmission:** While revising your submission, we strongly recommend that you use PLOS’s NAAS tool (https://ngplosjournals.pagemajik.ai/artanalysis) to test your figure files. NAAS can convert your figure files to the TIFF file type and meet basic requirements (such as print size, resolution), or provide you with a report on issues that do not meet our requirements and that NAAS cannot fix.

After uploading your figures to PLOS’s NAAS tool - https://ngplosjournals.pagemajik.ai/artanalysis, NAAS will process the files provided and display the results in the "Uploaded Files" section of the page as the processing is complete. If the uploaded figures meet our requirements (or NAAS is able to fix the files to meet our requirements), the figure will be marked as "fixed" above. If NAAS is unable to fix the files, a red "failed" label will appear above. When NAAS has confirmed that the figure files meet our requirements, please download the file via the download option, and include these NAAS processed figure files when submitting your revised manuscript. **Reproducibility:** To enhance the reproducibility of your results, we recommend that authors of applicable studies deposit laboratory protocols in protocols.io, where a protocol can be assigned its own identifier (DOI) such that it can be cited independently in the future. Additionally, PLOS ONE offers an option to publish peer-reviewed clinical study protocols. Read more information on sharing protocols at https://plos.org/protocols?utm_medium=editorial-email&utm_source=authorletters&utm_campaign=protocols

---

## [Decision Letter · Decision Letter 1]

6 Mar 2026

The MARC SE-Africa Dashboard: Joining Forces to Counteract Emerging Antimalarial Resistance in South and East Africa

PDIG-D-25-00012R1

Dear Dr. van Wyk,

We are pleased to inform you that your manuscript 'The MARC SE-Africa Dashboard: Joining Forces to Counteract Emerging Antimalarial Resistance in South and East Africa' has been provisionally accepted for publication in PLOS Digital Health.

Best regards,

Louise A C Millard, PhD

Section Editor

PLOS Digital Health

**Additional Editor Comments (if provided):**

**Reviewer Comments (if any, and for reference):**

Reviewer's Responses to Questions

**Comments to the Author**

1. If the authors have adequately addressed your comments raised in a previous round of review and you feel that this manuscript is now acceptable for publication, you may indicate that here to bypass the “Comments to the Author” section, enter your conflict of interest statement in the “Confidential to Editor” section, and submit your "Accept" recommendation.

Reviewer #1: All comments have been addressed

2. Does this manuscript meet PLOS Digital Health’s publication criteria? Is the manuscript technically sound, and do the data support the conclusions? The manuscript must describe methodologically and ethically rigorous research with conclusions that are appropriately drawn based on the data presented.? Is the manuscript technically sound, and do the data support the conclusions? The manuscript must describe methodologically and ethically rigorous research with conclusions that are appropriately drawn based on the data presented.

Reviewer #1: Yes

3. Has the statistical analysis been performed appropriately and rigorously?

Reviewer #1: Yes

4. Have the authors made all data underlying the findings in their manuscript fully available (please refer to the Data Availability Statement at the start of the manuscript PDF file)?

The PLOS Data policy requires authors to make all data underlying the findings described in their manuscript fully available without restriction, with rare exception. The data should be provided as part of the manuscript or its supporting information, or deposited to a public repository. For example, in addition to summary statistics, the data points behind means, medians and variance measures should be available. If there are restrictions on publicly sharing data—e.g. participant privacy or use of data from a third party—those must be specified.requires authors to make all data underlying the findings described in their manuscript fully available without restriction, with rare exception. The data should be provided as part of the manuscript or its supporting information, or deposited to a public repository. For example, in addition to summary statistics, the data points behind means, medians and variance measures should be available. If there are restrictions on publicly sharing data—e.g. participant privacy or use of data from a third party—those must be specified.

Reviewer #1: Yes

5. Is the manuscript presented in an intelligible fashion and written in standard English?

Reviewer #1: Yes

6. Review Comments to the Author

Reviewer #1: The authors have adequately revised the manuscript in accordance with my previous suggestions. The language has been notably improved, and the limitations of the study have been appropriately addressed. I have no further concerns.

7. PLOS authors have the option to publish the peer review history of their article (what does this mean?). If published, this will include your full peer review and any attached files.). If published, this will include your full peer review and any attached files.

**Do you want your identity to be public for this peer review?** For information about this choice, including consent withdrawal, please see our Privacy Policy..

Reviewer #1: No
